# A Note on the Convergence of Denoising Diffusion Probabilistic Models

**Sokhna Diarra Mbacke**               *sokhna-diarra.mbacke.1@ulaval.ca*
*Université Laval*

**Omar Rivasplata**                   *o.rivasplata@ucl.ac.uk*
*University College London*

**Reviewed on OpenReview:** *https://openreview.net/forum?id=wLe1bG93yc*

## Abstract

Diffusion models are one of the most important families of deep generative models. In this note, we derive a quantitative upper bound on the Wasserstein distance between the target distribution and the distribution learned by a diffusion model. Unlike previous works on this topic, our result does not make assumptions on the learned score function. Moreover, our result holds for arbitrary data-generating distributions on bounded instance spaces, even those without a density with respect to Lebesgue measure, and the upper bound does not suffer from exponential dependencies on the ambient space dimension. Our main result builds upon the recent work of Mbacke et al. (2023) and our proofs are elementary.

## 1 Introduction

Along with generative adversarial networks (Goodfellow et al., 2014) and variational autoencoders (VAEs) (Kingma & Welling, 2014; Rezende et al., 2014), diffusion models (Sohl-Dickstein et al., 2015; Song & Ermon, 2019; Ho et al., 2020) are one of the most prominent families of deep generative models. They have exhibited impressive empirical performance in image (Dhariwal & Nichol, 2021; Ho et al., 2022) and audio (Chen et al., 2021; Popov et al., 2021) generation, as well as other applications (Zhou et al., 2021; Sasaki et al., 2021; Li et al., 2022; Trabucco et al., 2023).

There are two main approaches to diffusion models: denoising diffusion probabilistic models (DDPMs) (Sohl-Dickstein et al., 2015; Ho et al., 2020) and score-based generative models (Song & Ermon, 2019) (SGMs). The former kind, DDPMs, progressively transform samples from the target distribution into noise through a forward process, and train a backward process that reverses the transformation and is used to generate new samples. On the other hand, SGMs use score matching techniques (Hyvärinen & Dayan, 2005; Vincent, 2011) to learn an approximation of the score function of the data-generating distribution, then generate new samples using Langevin dynamics. Since for real-world distributions the score function might not exist, Song & Ermon (2019) propose adding different noise levels to the training samples to cover the whole instance space, and train a neural network to simultaneously learn the score function for all noise levels.

Although DDPMs and SGMs might appear to be different approaches at first, Ho et al. (2020) showed that DDPMs implicitly learn an approximation of the score function and the sampling process resembles Langevin dynamics. Furthermore, Song et al. (2021b) derived a unifying view of both techniques using stochastic differential equations (SDEs). The SGM of Song & Ermon (2019) can be seen as a discretization of the Brownian motion process, and the DDPM of Ho et al. (2020) as a discretization of an Ornstein–Uhlenbeck process. Hence, both DDPMs and SGMs are usually referred to as SGMs in the literature. This explains why the previous works studying the theoretical properties of diffusion models utilize the score-based formulation, which requires assumptions on the performance of the learned score function.

In this work, we take a different approach and apply techniques developed by Mbacke et al. (2023) for VAEs to DDPMs, which can be seen as hierarchical VAEs with fixed encoders (Luo, 2022). This approach allows us to derive quantitative Wasserstein-based upper bounds, with no assumptions on the data distribution, no assumptions on the learned score function, and elementary proofs that do not require the SDE toolbox. Moreover, our bounds do not suffer from any costly discretization step, such as the one in De Bortoli (2022), since we consider the forward and backward processes as being discrete-time from the outset, instead of seeing them as discretizations of continuous-time processes.

## 1.1 Related Works

There has been a growing body of work aiming to establish theoretical results on the convergence of SGMs (Block et al., 2020; De Bortoli et al., 2021; Song et al., 2021a; Lee et al., 2022; De Bortoli, 2022; Kwon et al., 2022; Lee et al., 2023; Chen et al., 2023; Li et al., 2023; Benton et al., 2023), but these works either rely on strong assumptions on the data-generating distribution, derive non quantitative upper bounds, or suffer from exponential dependencies on some of the parameters. We manage to avoid all three of these pitfalls. The bounds of Lee et al. (2022) rely on very strong assumptions on the data-generating distribution, such as log-Sobolev inequalities, which are not realistic for real-world data distributions. Furthermore, Song et al. (2021a); Chen et al. (2023); Lee et al. (2023) establish upper bounds on the Kullback-Leibler (KL) divergence or the total variation (TV) distance between the data-generating distribution and the distribution learned by the diffusion model; however, as noted by Pidstrigach (2022) and Chen et al. (2023), unless one makes strong assumptions on the support of the data-generating distribution, KL and TV reach their maximum values. Such assumptions arguably do not hold for real-world data-generating distributions which are widely believed to satisfy the manifold hypothesis (Narayanan & Mitter, 2010; Fefferman et al., 2016; Pope et al., 2021). The work of Pidstrigach (2022) establishes conditions under which the support of the input distribution is equal to the support of the learned distribution, and generalizes the bound of Song et al. (2021a) to all $f$-divergences. Assuming $L_2$ accurate score estimation, Chen et al. (2023) and Lee et al. (2023) establish Wasserstein distance upper bounds under weaker assumptions on the data-generating distribution, but their Wasserstein-based bounds are not quantitative. De Bortoli (2022) derives quantitative Wasserstein distance upper bounds under the manifold hypothesis, but their bounds suffer from exponential dependencies on some of the problem parameters.

## 1.2 Our contributions

In this work, we avoid strong assumptions on the data-generating distribution, and establish a quantitative Wasserstein distance upper bound without exponential dependencies on problem parameters including ambient space dimension. Moreover, a common thread in the works cited above is that their bounds depend on the error of the score estimator. According to Chen et al. (2023), "Providing precise guarantees for estimation of the score function is difficult, as it requires an understanding of the non-convex training dynamics of neural network optimization that is currently out of reach." Hence, we derive upper bounds without assumptions on the learned score function. Instead, our bound depends on a *reconstruction loss* computed on a finite i.i.d. sample. Intuitively, we define a loss function $\ell^\theta(\mathbf{x}_T, \mathbf{x}_0)$ (see equation 6), which measures the average Euclidean distance between a sample $\mathbf{x}_0$ from the data-generating distribution, and the reconstruction $\hat{\mathbf{x}}_0$ obtained by sampling noise $\mathbf{x}_T \sim q(\mathbf{x}_T|\mathbf{x}_0)$ and passing it through the backward process (parameterized by $\theta$). This approach is motivated by the work of Mbacke et al. (2023) on VAEs.

There are many advantages to this approach: no restrictive assumptions on the data-generating distribution, no exponential dependencies on the dimension, and a quantitative upper bound based on the Wasserstein distance. Moreover, our approach has the benefit of utilizing very simple and elementary proofs.

## 2 Preliminaries

Throughout the paper, we use lower-case letters to denote both probability measures and their densities w.r.t. the Lebesgue measure, and we add variables in parentheses to improve readability (e.g. $q(\mathbf{x}_t|\mathbf{x}_{t-1})$ to indicate a time-dependent conditional distribution). We consider an instance space $\mathcal{X}$ which is a subset of $\mathbb{R}^D$ with

Figure 1: Denoising diffusion model

the Euclidean distance as underlying metric, and a target data-generating distribution $\mu \in \mathcal{M}_+^1(\mathcal{X})$. Notice that we do not assume $\mu$ has a density w.r.t. the Lebesgue measure. Moreover, $\|\cdot\|$ denotes the Euclidean ($L_2$) norm and we write $\mathbb{E}_{p(\mathbf{x})}$ as a shorthand for $\mathbb{E}_{\mathbf{x} \sim p(\mathbf{x})}$. Given probability measures $p, q \in \mathcal{M}_+^1(\mathcal{X})$ and a real number $k > 1$, the Wasserstein distance of order $k$ is defined as (Villani, 2009):

$$W_k(p,q) = \left( \inf_{\pi \in \Gamma(p,q)} \int_{\mathcal{X}} \|\mathbf{x} - \mathbf{y}\|^k \, d\pi(\mathbf{x}, \mathbf{y}) \right)^{1/k},$$

where $\Gamma(p,q)$ denotes the set of couplings of $p$ and $q$, meaning the set of joint distributions on $\mathcal{X} \times \mathcal{X}$ with respective marginals $p$ and $q$. We refer to the product measure $p \otimes q$ as the *trivial coupling*, and we refer to the Wasserstein distance of order 1 simply as *the Wasserstein distance*.

## 2.1 Denoising Diffusion Models

Instead of using the SDE arsenal, we present diffusion models using the DDPM formulation with discrete-time processes. A diffusion model comprises two discrete-time stochastic processes: a forward process and a backward process. Both processes are indexed by time $0 \leq t \leq T$, where the number of time-steps $T$ is a pre-set choice. See Figure 1 for an illustration, and Luo (2022) for a detailed tutorial.

**The forward process.** The forward process transforms a datapoint $\mathbf{x}_0 \sim \mu$ into a noise distribution $q(\mathbf{x}_T | \mathbf{x}_0)$, via a sequence of conditional distributions $q(\mathbf{x}_t | \mathbf{x}_{t-1})$ for $1 \leq t \leq T$. It is assumed that the forward process is defined so that for large enough $T$, the distribution $q(\mathbf{x}_T | \mathbf{x}_0)$ is close to a simple noise distribution $p(\mathbf{x}_T)$ which is referred to as the *prior distribution*. For instance, Ho et al. (2020) chose $p(\mathbf{x}_T) = \mathcal{N}(\mathbf{x}_T; \mathbf{0}, \mathbf{I})$, the standard multivariate normal distribution.

**The backward process.** The backward process is a Markov process with parametric transition kernels. The goal of the backward process is to implement the reverse action of the forward process: transforming noise samples into (approximate) samples from the distribution $\mu$. Following Ho et al. (2020), we assume the backward process to be defined by Gaussian distributions $p_\theta(\mathbf{x}_{t-1} | \mathbf{x}_t)$ defined for $2 \leq t \leq T$ as

$$p_\theta(\mathbf{x}_{t-1} | \mathbf{x}_t) = \mathcal{N}\left(\mathbf{x}_{t-1}; g_\theta^t(\mathbf{x}_t), \sigma_t^2 \mathbf{I}\right), \tag{1}$$

and

$$p_\theta(\mathbf{x}_0 | \mathbf{x}_1) = g_\theta^1(\mathbf{x}_1), \tag{2}$$

where the variance parameters $\sigma_1^2, \ldots, \sigma_T^2 \in \mathbb{R}_{\geq 0}$ are defined by a fixed schedule, the mean functions $g_\theta^t : \mathbb{R}^D \to \mathbb{R}^D$ are learned using a neural network (with parameters $\theta$) for $2 \leq t \leq T$, and $g_\theta^1 : \mathbb{R}^D \to \mathcal{X}$ is a separate function dependent on $\sigma_1$. In practice, Ho et al. (2020) used the same network for the functions $g_\theta^t$ for $2 \leq t \leq T$, and a separate discrete decoder for $g_\theta^1$.

Generating new samples from a trained diffusion model is done by sampling $\mathbf{x}_{t-1} \sim p_\theta(\mathbf{x}_{t-1} | \mathbf{x}_t)$ for $1 \leq t \leq T$, starting from a noise vector $\mathbf{x}_T \sim p(\mathbf{x}_T)$ sampled from the prior $p(\mathbf{x}_T)$.

We make the following assumption on the backward process.

**Assumption 1.** We assume for each $1 \leq t \leq T$ there exists a constant $K_\theta^t > 0$ such that for every $\mathbf{x}_1, \mathbf{x}_2 \in \mathcal{X}$,

$$\left\| g_\theta^t(\mathbf{x}_1) - g_\theta^t(\mathbf{x}_2) \right\| \leq K_\theta^t \left\| \mathbf{x}_1 - \mathbf{x}_2 \right\|.$$

In other words, $g_\theta^t$ is $K_\theta^t$-Lipschitz continuous. We discuss this assumption in Remark 3.2.

## 2.2 Additional Definitions

We define the distribution $\pi_\theta(\cdot|\mathbf{x}_0)$ as

$$\pi_\theta(\cdot|\mathbf{x}_0) = q(\mathbf{x}_T|\mathbf{x}_0)p_\theta(\mathbf{x}_{T-1}|\mathbf{x}_T)p_\theta(\mathbf{x}_{T-2}|\mathbf{x}_{T-1})\ldots p_\theta(\mathbf{x}_1|\mathbf{x}_2)p_\theta(\cdot|\mathbf{x}_1). \tag{3}$$

Intuitively, for each $\mathbf{x}_0 \in \mathcal{X}$, $\pi_\theta(\cdot|\mathbf{x}_0)$ denotes the distribution on $\mathcal{X}$ obtained by reconstructing samples from $q(\mathbf{x}_T|\mathbf{x}_0)$ through the backward process. Another way of seeing this distribution is that for any function $f : \mathcal{X} \to \mathbb{R}$, the following equation holds:[1]

$$\underset{\pi_\theta(\hat{\mathbf{x}}_0|\mathbf{x}_0)}{\mathbb{E}} f(\hat{\mathbf{x}}_0) = \underset{q(\mathbf{x}_T|\mathbf{x}_0)}{\mathbb{E}} \underset{p_\theta(\mathbf{x}_{T-1}|\mathbf{x}_T)}{\mathbb{E}} \cdots \underset{p_\theta(\mathbf{x}_1|\mathbf{x}_2)}{\mathbb{E}} \underset{p_\theta(\hat{\mathbf{x}}_0|\mathbf{x}_1)}{\mathbb{E}} f(\hat{\mathbf{x}}_0). \tag{4}$$

Given a finite set $S = \{\mathbf{x}_0^1, \ldots, \mathbf{x}_0^n\} \overset{\text{iid}}{\sim} \mu$, we define the regenerated distribution as the following mixture:

$$\mu_\theta^n = \frac{1}{n} \sum_{i=1}^n \pi_\theta(\cdot|\mathbf{x}_0^i). \tag{5}$$

This definition is analogous to the empirical regenerated distribution defined by Mbacke et al. (2023) for VAEs. The distribution on $\mathcal{X}$ learned by the diffusion model is denoted $\pi_\theta(\cdot)$ and defined as

$$\pi_\theta(\cdot) = p(\mathbf{x}_T)p_\theta(\mathbf{x}_{T-1}|\mathbf{x}_T)p_\theta(\mathbf{x}_{T-2}|\mathbf{x}_{T-1})\ldots p_\theta(\mathbf{x}_1|\mathbf{x}_2)p_\theta(\cdot|\mathbf{x}_1).$$

In other words, for any function $f : \mathcal{X} \to \mathbb{R}$, the expectation of $f$ w.r.t. $\pi_\theta(\cdot)$ is

$$\underset{\pi_\theta(\hat{\mathbf{x}}_0)}{\mathbb{E}} f(\hat{\mathbf{x}}_0) = \underset{p(\mathbf{x}_T)}{\mathbb{E}} \underset{p_\theta(\mathbf{x}_{T-1}|\mathbf{x}_T)}{\mathbb{E}} \cdots \underset{p_\theta(\mathbf{x}_1|\mathbf{x}_2)}{\mathbb{E}} \underset{p_\theta(\hat{\mathbf{x}}_0|\mathbf{x}_1)}{\mathbb{E}} f(\hat{\mathbf{x}}_0).$$

Hence, both $\pi_\theta(\cdot)$ and $\pi_\theta(\cdot|\mathbf{x}_0)$ are defined using the backward process, with the difference that $\pi_\theta(\cdot)$ starts with the prior $p(\mathbf{x}_T) = \mathcal{N}(\mathbf{x}_T; \mathbf{0}, \mathbf{I})$ while $\pi_\theta(\cdot|\mathbf{x}_0)$ starts with the noise distribution $q(\mathbf{x}_T|\mathbf{x}_0)$.

Finally, we define the loss function $\ell^\theta : \mathcal{X} \times \mathcal{X} \to \mathbb{R}$ as

$$\ell^\theta(\mathbf{x}_T, \mathbf{x}_0) = \underset{p_\theta(\mathbf{x}_{T-1}|\mathbf{x}_T)}{\mathbb{E}} \underset{p_\theta(\mathbf{x}_{T-2}|\mathbf{x}_{T-1})}{\mathbb{E}} \cdots \underset{p_\theta(\mathbf{x}_1|\mathbf{x}_2)}{\mathbb{E}} \underset{p_\theta(\hat{\mathbf{x}}_0|\mathbf{x}_1)}{\mathbb{E}} \|\mathbf{x}_0 - \hat{\mathbf{x}}_0\|. \tag{6}$$

Hence, given a noise vector $\mathbf{x}_T$ and a sample $\mathbf{x}_0$, the loss $\ell^\theta(\mathbf{x}_T, \mathbf{x}_0)$ denotes the average Euclidean distance between $\mathbf{x}_0$ and any sample obtained by passing $\mathbf{x}_T$ through the backward process.

## 2.3 Our Approach

The goal is to upper-bound the distance $W_1(\mu, \pi_\theta(\cdot))$. Since the triangle inequality implies

$$W_1(\mu, \pi_\theta(\cdot)) \leq W_1(\mu, \mu_\theta^n) + W_1(\mu_\theta^n, \pi_\theta(\cdot)), \tag{7}$$

we can upper-bound the distance $W_1(\mu, \pi_\theta(\cdot))$ by upper-bounding the two expressions on the right-hand side of equation 7 separately. The upper bound on $W_1(\mu, \mu_\theta^n)$ is obtained using a straightforward adaptation of a proof that was developed by Mbacke et al. (2023). First, $W_1(\mu, \mu_\theta^n)$ is upper-bounded using the expectation of the loss function $\ell^\theta$, then the resulting expression is upper-bounded using a PAC-Bayesian-style expression dependent on the empirical risk and the prior-matching term.

The upper bound on the second term $W_1(\mu_\theta^n, \pi_\theta(\cdot))$ uses the definition of $\mu_\theta^n$. Intuitively, the difference between $\pi_\theta(\cdot|\mathbf{x}_0^i)$ and $\pi_\theta(\cdot)$ is determined by the corresponding initial distributions: $q(\mathbf{x}_T|\mathbf{x}_0^i)$ for $\pi_\theta(\cdot|\mathbf{x}_0^i)$, and $p(\mathbf{x}_T)$ for $\pi_\theta(\cdot)$. Hence, if the two initial distributions are close, and if the steps of the backward process are smooth (see Assumption 1), then $\pi_\theta(\cdot|\mathbf{x}_0^i)$ and $\pi_\theta(\cdot)$ are close to each other.

---

[1]More formally, we give a definition of $\pi_\theta(\cdot|\mathbf{x}_0)$ via expectations of test functions by requiring that equation 4 holds for every function $f$ in some appropriate measure-determining function class.

# 3 Main Result

## 3.1 Theorem Statement

We are now ready to state our main result: a quantitative upper bound on the Wasserstein distance between the data-generating distribution $\mu$ and the learned distribution $\pi_\theta(\cdot)$.

**Theorem 3.1.** *Assume the instance space $\mathcal{X}$ has finite diameter $\Delta = \sup_{\mathbf{x}, \mathbf{x}' \in \mathcal{X}} \|\mathbf{x} - \mathbf{x}'\| < \infty$, and let $\lambda > 0$ and $\delta \in (0, 1)$ be real numbers. Using the definitions and assumptions of the previous section, the following inequality holds with probability at least $1 - \delta$ over the random draw of $S = \{\mathbf{x}_0^1, \ldots, \mathbf{x}_0^n\} \overset{iid}{\sim} \mu$:*

$$
\begin{aligned}
W_1(\mu, \pi_\theta(\cdot)) \leq \frac{1}{n} \sum_{i=1}^{n} \left\{ \underset{q(\mathbf{x}_T|\mathbf{x}_0^i)}{\mathbb{E}} \ell^\theta(\mathbf{x}_T, \mathbf{x}_0^i) \right\} + \frac{1}{\lambda} \left[ \sum_{i=1}^{n} \text{KL}(q(\mathbf{x}_T|\mathbf{x}_0^i) \,\|\, p(\mathbf{x}_T)) + \log \frac{1}{\delta} \right] + \frac{\lambda \Delta^2}{8n} + \\
\left( \prod_{t=1}^{T} K_\theta^t \right) \frac{1}{n} \sum_{i=1}^{n} \left\{ \underset{q(\mathbf{x}_T|\mathbf{x}_0^i)}{\mathbb{E}} \underset{p(\mathbf{y}_T)}{\mathbb{E}} \|\mathbf{x}_T - \mathbf{y}_T\| \right\} + \left( \sum_{t=2}^{T} \left( \prod_{i=1}^{t-1} K_\theta^i \right) \sigma_t \right) \underset{\epsilon, \epsilon'}{\mathbb{E}} \|\boldsymbol{\epsilon} - \boldsymbol{\epsilon}'\|.
\end{aligned}
\tag{8}
$$

*Where $\boldsymbol{\epsilon}, \boldsymbol{\epsilon}' \sim \mathcal{N}(\mathbf{0}, \mathbf{I})$ are standard Gaussian vectors.*

**Remark 3.1.** Before presenting the proof, let us discuss Theorem 3.1.

- Because the right-hand side of equation 8 depends on a quantity computed using a finite i.i.d. sample $S$, the bound holds with high probability w.r.t. the randomness of $S$. This is the price we pay for having a quantitative upper bound with no exponential dependencies on problem parameters and no assumptions on the data-generating distribution $\mu$.

- The first term of the right-hand side of equation 8 is the average reconstruction loss computed over the sample $S = \{\mathbf{x}_0^1, \ldots, \mathbf{x}_0^n\}$. Note that for each $1 \leq i \leq n$, the expectation of $\ell^\theta(\mathbf{x}_T|\mathbf{x}_0^i)$ is only computed w.r.t. the noise distribution $q(\mathbf{x}_T|\mathbf{x}_0^i)$ defined by $\mathbf{x}_0^i$ itself. Hence, this term measures how well a noise vector $\mathbf{x}_T \sim q(\mathbf{x}_T|\mathbf{x}_0^i)$ recovers the original sample $\mathbf{x}_0^i$ using the backward process, and averages over the set $S = \{\mathbf{x}_0^1, \ldots, \mathbf{x}_0^n\}$.

- If the Lipschitz constants satisfy $K_\theta^t < 1$ for all $1 \leq t \leq T$, then the larger $T$ is, the smaller the upper bound gets. This is because the product of $K_\theta^t$'s then converges to 0. In Remark 3.2 below, we show that the assumption that $K_\theta^t < 1$ for all $t$ is a quite reasonable one.

- The hyperparameter $\lambda$ controls the trade-off between the prior-matching (KL) term and the diameter term $\frac{\Delta^2}{8n}$. If $K_\theta^t < 1$ for all $1 \leq t \leq T$ and $T \to \infty$, then the convergence of the bound largely depends on the choice of $\lambda$. In that case, $\lambda \propto n^{1/2}$ leads to a faster convergence, while $\lambda \propto n$ leads to a slower convergence to a smaller quantity. This is because the bound of Mbacke et al. (2023) stems from PAC-Bayesian theory, where this trade-off is common, see e.g. Alquier (2021).

- The last term of equation 8 does not depend on the sample size $n$. Hence, the upper bound given by Theorem 3.1 does not converge to 0 as $n \to \infty$. However, if the Lipschitz factors $(K_\theta^t)_{1 \leq t \leq T}$ are all less than 1, then this term can be very small, specially in low dimensional spaces.

## 3.2 Proof of the main theorem

The following result is an adaptation of a result by Mbacke et al. (2023).

**Lemma 3.2.** *Let $\lambda > 0$ and $\delta \in (0, 1)$ be real numbers. With probability at least $1 - \delta$ over the randomness of the sample $S = \{\mathbf{x}_0^1, \ldots, \mathbf{x}_0^n\} \overset{iid}{\sim} \mu$, the following holds:*

$$
W_1(\mu, \mu_\theta^n) \leq \frac{1}{n} \sum_{i=1}^{n} \left\{ \underset{q(\mathbf{x}_T|\mathbf{x}_0^i)}{\mathbb{E}} \ell^\theta(\mathbf{x}_T, \mathbf{x}_0^i) \right\} + \frac{1}{\lambda} \left[ \sum_{i=1}^{n} \text{KL}(q(\mathbf{x}_T|\mathbf{x}_0^i) \,\|\, p(\mathbf{x}_T)) + \log \frac{1}{\delta} \right] + \frac{\lambda \Delta^2}{8n}.
\tag{9}
$$

The proof of this result is a straightforward adaptation of Mbacke et al. (2023, Lemma D.1). We provide the proof in the supplementary material (Section A.1) for completeness.

Now, let us focus our attention on the second term of the right-hand side of equation 7, namely $W_1(\mu_\theta^n, \pi_\theta(\cdot))$. This part is trickier than for VAEs, for which the generative model's distribution is simply a pushforward measure. Here, we have a non-deterministic sampling process with $T$ steps.

Assumption 1 leads to the following lemma on the backward process.

**Lemma 3.3.** *For any given* $\mathbf{x}_1, \mathbf{y}_1 \in \mathcal{X}$ *we have*

$$\mathbb{E}_{p_\theta(\mathbf{x}_0|\mathbf{x}_1)} \mathbb{E}_{p_\theta(\mathbf{y}_0|\mathbf{y}_1)} \|\mathbf{x}_0 - \mathbf{y}_0\| \leq K_\theta^1 \|\mathbf{x}_1 - \mathbf{y}_1\|.$$

*Moreover, if* $2 \leq t \leq T$, *then for any given* $\mathbf{x}_t, \mathbf{y}_t \in \mathcal{X}$ *we have*

$$\mathbb{E}_{p_\theta(\mathbf{x}_{t-1}|\mathbf{x}_t)} \mathbb{E}_{p_\theta(\mathbf{y}_{t-1}|\mathbf{y}_t)} \|\mathbf{x}_{t-1} - \mathbf{y}_{t-1}\| \leq K_\theta^t \|\mathbf{x}_t - \mathbf{y}_t\| + \sigma_t \mathbb{E}_{\boldsymbol{\epsilon},\boldsymbol{\epsilon}'} \|\boldsymbol{\epsilon} - \boldsymbol{\epsilon}'\|,$$

*where* $\boldsymbol{\epsilon}, \boldsymbol{\epsilon}' \sim \mathcal{N}(\mathbf{0}, \mathbf{I})$, *meaning* $\mathbb{E}_{\boldsymbol{\epsilon},\boldsymbol{\epsilon}'}$ *is a shorthand for* $\mathbb{E}_{\boldsymbol{\epsilon},\boldsymbol{\epsilon}' \sim \mathcal{N}(\mathbf{0}, \mathbf{I})}$.

*Proof.* For the first part, let $\mathbf{x}_1, \mathbf{y}_1 \in \mathcal{X}$. Since according to equation 2 we have $p_\theta(\mathbf{x}_0|\mathbf{x}_1) = \delta_{g_\theta^1(\mathbf{x}_1)}(\mathbf{x}_0)$ and $p_\theta(\mathbf{y}_0|\mathbf{y}_1) = \delta_{g_\theta^1(\mathbf{y}_1)}(\mathbf{y}_0)$, then

$$\mathbb{E}_{p_\theta(\mathbf{x}_0|\mathbf{x}_1)} \mathbb{E}_{p_\theta(\mathbf{y}_0|\mathbf{y}_1)} \|\mathbf{x}_0 - \mathbf{y}_0\| = \left\| g_\theta^1(\mathbf{x}_1) - g_\theta^1(\mathbf{y}_1) \right\| \leq K_\theta^1 \|\mathbf{x}_1 - \mathbf{y}_1\|.$$

For the second part, let $2 \leq t \leq T$ and $\mathbf{x}_t, \mathbf{y}_t \in \mathcal{X}$. Since $p_\theta(\mathbf{x}_{t-1}|\mathbf{x}_t) = \mathcal{N}\left(\mathbf{x}_{t-1}; g_\theta^t(\mathbf{x}_t), \sigma_t^2 \mathbf{I}\right)$ by equation 1, the reparameterization trick (Kingma & Welling, 2014) implies that sampling

$$\mathbf{x}_{t-1} \sim p_\theta(\mathbf{x}_{t-1}|\mathbf{x}_t)$$

is equivalent to setting

$$\mathbf{x}_{t-1} = g_\theta^t(\mathbf{x}_t) + \sigma_t \boldsymbol{\epsilon}_t, \quad \text{with} \quad \boldsymbol{\epsilon}_t \sim \mathcal{N}(\mathbf{0}, \mathbf{I}). \tag{10}$$

Using equation 10, the triangle inequality, and Assumption 1, we obtain

$$\begin{aligned}
\mathbb{E}_{p_\theta(\mathbf{x}_{t-1}|\mathbf{x}_t)} \mathbb{E}_{p_\theta(\mathbf{y}_{t-1}|\mathbf{y}_t)} \|\mathbf{x}_{t-1} - \mathbf{y}_{t-1}\| &= \mathbb{E}_{\boldsymbol{\epsilon}_t} \mathbb{E}_{\boldsymbol{\epsilon}_t'} \left\| g_\theta^t(\mathbf{x}_t) + \sigma_t \boldsymbol{\epsilon}_t - g_\theta^t(\mathbf{y}_t) - \sigma_t \boldsymbol{\epsilon}_t' \right\| \\
&\leq \mathbb{E}_{\boldsymbol{\epsilon}_t} \mathbb{E}_{\boldsymbol{\epsilon}_t'} \left\| g_\theta^t(\mathbf{x}_t) - g_\theta^t(\mathbf{y}_t) \right\| + \sigma_t \mathbb{E}_{\boldsymbol{\epsilon}_t} \mathbb{E}_{\boldsymbol{\epsilon}_t'} \|\boldsymbol{\epsilon}_t - \boldsymbol{\epsilon}_t'\| \\
&= \left\| g_\theta^t(\mathbf{x}_t) - g_\theta^t(\mathbf{y}_t) \right\| + \sigma_t \mathbb{E}_{\boldsymbol{\epsilon}_t} \mathbb{E}_{\boldsymbol{\epsilon}_t'} \|\boldsymbol{\epsilon}_t - \boldsymbol{\epsilon}_t'\| \\
&\leq K_\theta^t \|\mathbf{x}_t - \mathbf{y}_t\| + \sigma_t \mathbb{E}_{\boldsymbol{\epsilon}} \mathbb{E}_{\boldsymbol{\epsilon}'} \|\boldsymbol{\epsilon} - \boldsymbol{\epsilon}'\|,
\end{aligned}$$

where $\boldsymbol{\epsilon}, \boldsymbol{\epsilon}' \sim \mathcal{N}(\mathbf{0}, \mathbf{I})$. $\qquad\square$

Next, we can use the inequalities of Lemma 3.3 to prove the following result.

**Lemma 3.4.** *Let* $T \geq 1$. *The following inequality holds:*

$$\mathbb{E}_{p_\theta(\mathbf{x}_{T-1}|\mathbf{x}_T)} \mathbb{E}_{p_\theta(\mathbf{y}_{T-1}|\mathbf{y}_T)} \mathbb{E}_{p_\theta(\mathbf{x}_{T-2}|\mathbf{x}_{T-1})} \mathbb{E}_{p_\theta(\mathbf{y}_{T-2}|\mathbf{y}_{T-1})} \cdots \mathbb{E}_{p_\theta(\mathbf{x}_0|\mathbf{x}_1)} \mathbb{E}_{p_\theta(\mathbf{y}_0|\mathbf{y}_1)} \|\mathbf{x}_0 - \mathbf{y}_0\| \leq$$

$$\left(\prod_{t=1}^T K_\theta^t\right) \|\mathbf{x}_T - \mathbf{y}_T\| + \left(\sum_{t=2}^T \left(\prod_{i=1}^{t-1} K_\theta^i\right) \sigma_t\right) \mathbb{E}_{\boldsymbol{\epsilon},\boldsymbol{\epsilon}'} \left[\|\boldsymbol{\epsilon} - \boldsymbol{\epsilon}'\|\right],$$

*where* $\boldsymbol{\epsilon}, \boldsymbol{\epsilon}' \sim \mathcal{N}(\mathbf{0}, \mathbf{I})$.

*Proof Idea.* Lemma 3.4 is proven by induction using Lemma 3.3 in the induction step. The details are in the supplementary material (Section A.2). $\qquad\square$

Using the two previous lemmas, we obtain the following upper bound on $W_1(\mu_\theta^n, \pi_\theta(\cdot))$.

**Lemma 3.5.** *The following inequality holds:*

$$W_1(\mu_\theta^n, \pi_\theta(\cdot)) \leq \left(\prod_{t=1}^{T} K_\theta^t\right) \frac{1}{n} \sum_{i=1}^{n} \left\{ \underset{q(\mathbf{x}_T|\mathbf{x}_0^i)}{\mathbb{E}} \underset{p(\mathbf{y}_T)}{\mathbb{E}} \|\mathbf{x}_T - \mathbf{y}_T\| \right\} + \left(\sum_{t=2}^{T} \left(\prod_{i=1}^{t-1} K_\theta^i\right) \sigma_t\right) \underset{\boldsymbol{\epsilon},\boldsymbol{\epsilon}'}{\mathbb{E}} \|\boldsymbol{\epsilon} - \boldsymbol{\epsilon}'\|,$$

*where $\boldsymbol{\epsilon}, \boldsymbol{\epsilon}' \sim \mathcal{N}(\mathbf{0}, \mathbf{I})$.*

*Proof.* Using the definition of $W_1$, the trivial coupling, the definitions of $\mu_\theta^n$ and $\pi_\theta(\cdot)$, and Lemma 3.4, we get

$$
\begin{aligned}
W_1(\mu_\theta^n, \pi_\theta(\cdot)) &= \inf_{\pi \in \Gamma(\mu_\theta^n, \pi_\theta(\cdot))} \int d(\mathbf{x}, \mathbf{y}) \, d\pi(\mathbf{x}, \mathbf{y}) \\
&\leq \underset{\mathbf{x} \sim \mu_\theta^n}{\mathbb{E}} \underset{\mathbf{y} \sim \pi_\theta(\mathbf{y})}{\mathbb{E}} [\|\mathbf{x} - \mathbf{y}\|] \\
&= \frac{1}{n} \sum_{i=1}^{n} \left\{ \underset{q(\mathbf{x}_T|\mathbf{x}_0^i)}{\mathbb{E}} \underset{p(\mathbf{y}_T)}{\mathbb{E}} \underset{p_\theta(\mathbf{x}_{T-1}|\mathbf{x}_T)}{\mathbb{E}} \underset{p_\theta(\mathbf{y}_{T-1}|\mathbf{y}_T)}{\mathbb{E}} \cdots \underset{p_\theta(\mathbf{x}_0|\mathbf{x}_1)}{\mathbb{E}} \underset{p_\theta(\mathbf{y}_0|\mathbf{y}_1)}{\mathbb{E}} \|\mathbf{x}_0 - \mathbf{y}_0\| \right\} \\
&\leq \frac{1}{n} \sum_{i=1}^{n} \left\{ \underset{q(\mathbf{x}_T|\mathbf{x}_0^i)}{\mathbb{E}} \underset{p(\mathbf{y}_T)}{\mathbb{E}} \left[ \left(\prod_{t=1}^{T} K_\theta^t\right) \|\mathbf{x}_T - \mathbf{y}_T\| + \left(\sum_{t=2}^{T} \left(\prod_{j=1}^{t-1} K_\theta^j\right) \sigma_t\right) \underset{\boldsymbol{\epsilon},\boldsymbol{\epsilon}'}{\mathbb{E}} [\|\boldsymbol{\epsilon} - \boldsymbol{\epsilon}'\|] \right] \right\} \\
&= \left(\prod_{t=1}^{T} K_\theta^t\right) \frac{1}{n} \sum_{i=1}^{n} \left\{ \underset{q(\mathbf{x}_T|\mathbf{x}_0^i)}{\mathbb{E}} \underset{p(\mathbf{y}_T)}{\mathbb{E}} \|\mathbf{x}_T - \mathbf{y}_T\| \right\} + \left(\sum_{t=2}^{T} \left(\prod_{i=1}^{t-1} K_\theta^i\right) \sigma_t\right) \underset{\boldsymbol{\epsilon},\boldsymbol{\epsilon}'}{\mathbb{E}} \|\boldsymbol{\epsilon} - \boldsymbol{\epsilon}'\|.
\end{aligned}
$$

$\square$

Combining Lemmas 3.2 and 3.5 with equation 7 yields Theorem 3.1 .

### 3.3 Special case using the forward process of Ho et al. (2020)

Theorem 3.1 establishes a general upper bound that holds for any forward process, as long as the backward process satisfies Assumption 1. In this section, we specialize the statement of the theorem to the particular case of the forward process defined by Ho et al. (2020).

Let $\mathcal{X} \subseteq \mathbb{R}^D$. In Ho et al. (2020), the forward process is a Gauss-Markov process with transition densities defined as

$$q(\mathbf{x}_t|\mathbf{x}_{t-1}) = \mathcal{N}(\mathbf{x}_t; \sqrt{\alpha_t}\mathbf{x}_{t-1}, (1 - \alpha_t)\mathbf{I}),$$

where $\alpha_1, \ldots, \alpha_T$ is a fixed noise schedule such that $0 < \alpha_t < 1$ for all $t$. This definition implies that at each time step $1 \leq t \leq T$,

$$q(\mathbf{x}_t|\mathbf{x}_0) = \mathcal{N}(\mathbf{x}_t; \sqrt{\bar{\alpha}_t}\mathbf{x}_0, (1 - \bar{\alpha}_t)\mathbf{I}), \quad \text{with} \quad \bar{\alpha}_t = \prod_{i=1}^{t} \alpha_i.$$

The optimization objective to train the backward process ensures that for each time step $t$ the distribution $p_\theta(\mathbf{x}_{t-1}|\mathbf{x}_t)$ remains close to the ground-truth distribution $q(\mathbf{x}_{t-1}|\mathbf{x}_t, \mathbf{x}_0)$ given by

$$q(\mathbf{x}_{t-1}|\mathbf{x}_t, \mathbf{x}_0) = \mathcal{N}(\mathbf{x}_{t-1}; \mu_q^t(\mathbf{x}_t, \mathbf{x}_0), \sigma_t^2\mathbf{I}),$$

where

$$\mu_q^t(\mathbf{x}_t, \mathbf{x}_0) = \frac{\sqrt{\alpha_t}(1 - \bar{\alpha}_{t-1})}{1 - \bar{\alpha}_t}\mathbf{x}_t + \frac{\sqrt{\bar{\alpha}_{t-1}}(1 - \alpha_t)}{1 - \bar{\alpha}_t}\mathbf{x}_0. \tag{11}$$

Now, we discuss Assumption 1 under these definitions.

**Remark 3.2.** We can get a glimpse at the range of $K_\theta^t$ for a trained DDPM by looking at the distribution $q(\mathbf{x}_{t-1}|\mathbf{x}_t, \mathbf{x}_0)$, since $p_\theta(\mathbf{x}_{t-1}|\mathbf{x}_t)$ is optimized to be as close as possible to $q(\mathbf{x}_{t-1}|\mathbf{x}_t, \mathbf{x}_0)$.

For a given $\mathbf{x}_0 \sim \mu$, let us take a look at the Lipschitz norm of $\mathbf{x} \mapsto \mu_q^t(\mathbf{x}, \mathbf{x}_0)$. Using equation 11, we have

$$\mu_q^t(\mathbf{x}_t, \mathbf{x}_0) - \mu_q^t(\mathbf{y}_t, \mathbf{x}_0) = \frac{\sqrt{\alpha_t}(1 - \bar{\alpha}_{t-1})}{1 - \bar{\alpha}_t}(\mathbf{x}_t - \mathbf{y}_t).$$

Hence, $\mathbf{x} \mapsto \mu_q^t(\mathbf{x}, \mathbf{x}_0)$ is $K_t'$-Lipschitz continuous with

$$K_t' = \frac{\sqrt{\alpha_t}(1 - \bar{\alpha}_{t-1})}{1 - \bar{\alpha}_t}.$$

Now, if $\alpha_t < 1$ for all $1 \le t \le T$, then we have $1 - \bar{\alpha}_t > 1 - \bar{\alpha}_{t-1}$ which implies $K_t' < 1$ for all $1 \le t \le T$.

Remark 3.2 shows that the Lipschitz norm of the mean function $\mu_q^t(\cdot, \mathbf{x}_0)$ does not depend on $\mathbf{x}_0$. Indeed, looking at the previous equation, we can see that for any initial $\mathbf{x}_0$, the Lipschitz norm $K_t' = \frac{\sqrt{\alpha_t}(1-\bar{\alpha}_{t-1})}{1-\bar{\alpha}_t}$ only depends on the noise schedule, not $\mathbf{x}_0$ itself. Since $g_\theta^t$ is optimized to match to $\mu_q^t(\cdot, \mathbf{x}_0)$, for each $\mathbf{x}_0$ in the training set, and all the functions $\mu_q^t(\cdot, \mathbf{x}_0)$ have the same Lipschitz norm $K_t'$, we believe it is reasonable to assume $g_\theta^t$ is Lipschitz continuous as well. This is the intuition behind Assumption 1.

**The prior-matching term.** With the definitions of this section, the prior matching term $\mathrm{KL}(q(\mathbf{x}_T|\mathbf{x}_0) \| p(\mathbf{x}_T))$ has the following closed form:

$$\mathrm{KL}(q(\mathbf{x}_T|\mathbf{x}_0) \| p(\mathbf{x}_T)) = \frac{1}{2}\left[-D\log(1 - \bar{\alpha}_T) - D\bar{\alpha}_T + \bar{\alpha}_T \|\mathbf{x}_0\|^2\right].$$

**Upper-bounds on the average distance between Gaussian vectors.** If $\boldsymbol{\epsilon}, \boldsymbol{\epsilon}'$ are $D$-dimensional vectors sampled from $\mathcal{N}(\mathbf{0}, \mathbf{I})$, then

$$\mathbb{E}_{\boldsymbol{\epsilon}, \boldsymbol{\epsilon}'} \|\boldsymbol{\epsilon} - \boldsymbol{\epsilon}'\| \le \sqrt{2D}.$$

Moreover, since $q(\mathbf{x}_T|\mathbf{x}_0^i) = \mathcal{N}\left(\mathbf{x}_T; \bar{\alpha}_T \mathbf{x}_0^i, (1 - \bar{\alpha}_T)\mathbf{I}\right)$ and the prior $p(\mathbf{y}_T) = \mathcal{N}(\mathbf{y}_T; \mathbf{0}, \mathbf{I})$,

$$\mathbb{E}_{q(\mathbf{x}_T|\mathbf{x}_0^i)} \mathbb{E}_{p(\mathbf{y}_T)} \|\mathbf{x}_T - \mathbf{y}_T\| \le \sqrt{\bar{\alpha}_T \|\mathbf{x}_0^i\|^2 + (2 - \bar{\alpha}_T)D}.$$

**Special case of the main theorem.** With the definitions of this section, the inequality of Theorem 3.1 implies that with probability at least $1 - \delta$ over the randomness of $\{\mathbf{x}_0^1, \ldots, \mathbf{x}_0^n\} \overset{\text{iid}}{\sim} \mu$:

$$W_1(\mu, \pi_\theta(\cdot)) \le \frac{1}{n}\sum_{i=1}^n \left\{ \mathbb{E}_{q(\mathbf{x}_T|\mathbf{x}_0^i)} \ell^\theta(\mathbf{x}_T, \mathbf{x}_0^i) \right\} + \frac{1}{\lambda}\left[\frac{1}{2}\sum_{i=1}^n \left[-D\log(1 - \bar{\alpha}_T) - D\bar{\alpha}_T + \bar{\alpha}_T \|\mathbf{x}_0^i\|^2\right] + \log\frac{1}{\delta}\right] +$$

$$\frac{\lambda\Delta^2}{8n} + \left(\prod_{t=1}^T K_\theta^t\right)\frac{1}{n}\sum_{i=1}^n \sqrt{\bar{\alpha}_T \|\mathbf{x}_0^i\|^2 + (2 - \bar{\alpha}_T)D} + \sqrt{2D}\left(\sum_{t=2}^T\left(\prod_{i=1}^{t-1} K_\theta^i\right)\sigma_t\right).$$

## 4 Conclusion

This note presents a novel upper bound on the Wasserstein distance between the data-generating distribution and the distribution learned by a diffusion model. Unlike previous works in the field, our main result simultaneously avoids strong assumptions on the data-generating distribution, assumptions on the learned score function, and exponential dependencies, while still providing a quantitative upper bound. However, our bound holds with high probability on the randomness of a finite i.i.d. sample, on which a loss function is computed. Since the loss is a chain of expectations w.r.t. Gaussian distributions, it can either be estimated with high precision or upper bounded using the properties of Gaussian distributions.

## Acknowledgments

This research is supported by the Canada CIFAR AI Chair Program, and the NSERC Discovery grant RGPIN-2020- 07223. The authors sincerely thank Pascal Germain for interesting discussions and suggestions. The first author thanks Mathieu Bazinet and Florence Clerc for proof-reading the manuscript.

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

# A Omitted Proofs

## A.1 Proof of Lemma 3.2

Recall Lemma 3.2 states that the following inequality holds with probability $1 - \delta$:

$$W_1(\mu, \mu_\theta^n) \leq \frac{1}{n} \sum_{i=1}^n \left\{ \mathbb{E}_{q(\mathbf{x}_T|\mathbf{x}_0^i)} \ell^\theta(\mathbf{x}_T, \mathbf{x}_0^i) \right\} + \frac{1}{\lambda} \left[ \sum_{i=1}^n \mathrm{KL}(q(\mathbf{x}_T|\mathbf{x}_0^i) \| p(\mathbf{x}_T)) + \log \frac{1}{\delta} \right] + \frac{\lambda \Delta^2}{8n}.$$

*Proof of Lemma 3.2.* Using the trivial coupling (product of marginals), the definition of $\mu_\theta^n$ (equation 5), and the definition of the loss function $\ell^\theta$, we get

$$
\begin{aligned}
W_1(\mu, \mu_\theta^n) &= \inf_{\pi \in \Gamma(\mu, \mu_\theta^n)} \int_{\mathcal{X} \times \mathcal{X}} \|\mathbf{x} - \mathbf{y}\| \, d\pi(\mathbf{x}, \mathbf{y}) \\
&\leq \int_{\mathcal{X}} \int_{\mathcal{X}} \|\mathbf{x} - \mathbf{y}\| \, d\mu(\mathbf{x}) \, d\mu_\theta^n(\mathbf{y}) \\
&= \mathbb{E}_{\mathbf{x} \sim \mu} \mathbb{E}_{\mathbf{y} \sim \mu_\theta^n} \|\mathbf{x} - \mathbf{y}\| \\
&= \mathbb{E}_{\mathbf{x} \sim \mu} \left[ \frac{1}{n} \sum_{i=1}^n \left( \mathbb{E}_{q(\mathbf{x}_T|\mathbf{x}_0^i) \, p_\theta(\mathbf{x}_{T-1}|\mathbf{x}_T)} \cdots \mathbb{E}_{p_\theta(\mathbf{x}_1|\mathbf{x}_2) \, p_\theta(\mathbf{y}|\mathbf{x}_1)} \|\mathbf{x} - \mathbf{y}\| \right) \right] \\
&= \mathbb{E}_{\mathbf{x} \sim \mu} \left[ \frac{1}{n} \sum_{i=1}^n \mathbb{E}_{q(\mathbf{x}_T|\mathbf{x}_0^i)} \ell^\theta(\mathbf{x}_T, \mathbf{x}) \right].
\end{aligned}
$$

Using Mbacke et al. (2023, Lemma B.1), the following inequality holds with probability $1 - \delta$:

$$
\begin{aligned}
W_1(\mu, \mu_\theta^n) \leq{} & \frac{1}{n} \sum_{i=1}^n \left\{ \mathbb{E}_{q(\mathbf{x}_T|\mathbf{x}_0^i)} \ell^\theta(\mathbf{x}_T, \mathbf{x}_0^i) \right\} + \frac{1}{\lambda} \left[ \sum_{i=1}^n \mathrm{KL}(q(\mathbf{x}_T|\mathbf{x}_0^i) \| p(\mathbf{x}_T)) + \log \frac{1}{\delta} + \right. \\
& \left. n \log \mathbb{E}_{\mathbf{x}_0 \sim \mu^{\otimes n}} \mathbb{E}_{\mathbf{x}_T \sim p(\mathbf{x}_T)} e^{\lambda \left( \mathbb{E}_{\mathbf{y}_0 \sim \mu} [\ell^\theta(\mathbf{x}_T, \mathbf{y}_0)] - \frac{1}{n} \sum_{i=1}^n \ell^\theta(\mathbf{x}_T, \mathbf{x}_0)) \right)} \right].
\end{aligned}
\tag{12}
$$

Now, it remains to upper-bound the exponential moment of equation 9. If $\sup_{\mathbf{x}, \mathbf{x}' \in \mathcal{X}} \|\mathbf{x} - \mathbf{x}'\| = \Delta < \infty$, and $\lambda > 0$ is a real number, then the definition of the loss function $\ell^\theta$ and Hoeffding's lemma yield

$$n \log \mathbb{E}_{\mathbf{x}_0 \sim \mu} \mathbb{E}_{\mathbf{x}_T \sim p(\mathbf{x}_T)} e^{\lambda \left( \mathbb{E}_{\mathbf{x}' \sim \mu} [\ell^\theta(\mathbf{x}_T, \mathbf{x}')] - \frac{1}{n} \sum_{i=1}^n \ell^\theta(\mathbf{x}_T, \mathbf{x}_0)) \right)} \leq n \log \exp \left[ \frac{\lambda^2 \Delta^2}{8n^2} \right] = \frac{\lambda^2 \Delta^2}{8n}.$$

$\square$

## A.2 Proof of Lemma 3.4

Recall Lemma 3.4 states that the following inequality holds:

$$
\begin{aligned}
& \mathbb{E}_{p_\theta(\mathbf{x}_{T-1}|\mathbf{x}_T)} \mathbb{E}_{p_\theta(\mathbf{y}_{T-1}|\mathbf{y}_T)} \mathbb{E}_{p_\theta(\mathbf{x}_{T-2}|\mathbf{x}_{T-1})} \mathbb{E}_{p_\theta(\mathbf{y}_{T-2}|\mathbf{y}_{T-1})} \cdots \mathbb{E}_{p_\theta(\mathbf{x}_0|\mathbf{x}_1)} \mathbb{E}_{p_\theta(\mathbf{y}_0|\mathbf{y}_1)} \|\mathbf{x}_0 - \mathbf{y}_0\| \leq \\
& \qquad\qquad\qquad \left( \prod_{t=1}^T K_\theta^t \right) \|\mathbf{x}_T - \mathbf{y}_T\| + \left( \sum_{t=2}^T \left( \prod_{i=1}^{t-1} K_\theta^i \right) \sigma_t \right) \mathbb{E}_{\boldsymbol{\epsilon}, \boldsymbol{\epsilon}'} [\|\boldsymbol{\epsilon} - \boldsymbol{\epsilon}'\|],
\end{aligned}
$$

*Proof of Lemma 3.4.* Let's do a proof by induction on $T$.

- **Base case.** $T = 1$. The inequality becomes

$$\mathbb{E}_{p_\theta(\mathbf{x}_0|\mathbf{x}_1)} \mathbb{E}_{p_\theta(\mathbf{y}_0|\mathbf{y}_1)} \|\mathbf{x}_0 - \mathbf{y}_0\| \le K_\theta^1 \|\mathbf{x}_1 - \mathbf{y}_1\| \tag{13}$$

  which holds by the first part of Lemma 3.3.

- **Induction step.** Assume $T > 1$. The induction hypothesis is

$$\mathbb{E}_{p_\theta(\mathbf{x}_{T-2}|\mathbf{x}_{T-1}) p_\theta(\mathbf{y}_{T-2}|\mathbf{y}_{T-1})} \cdots \mathbb{E}_{p_\theta(\mathbf{x}_0|\mathbf{x}_1) p_\theta(\mathbf{y}_0|\mathbf{y}_1)} \|\mathbf{x}_0 - \mathbf{y}_0\| \le$$
$$\left(\prod_{t=1}^{T-1} K_\theta^t\right) \|\mathbf{x}_{T-1} - \mathbf{y}_{T-1}\| + \left(\sum_{t=2}^{T-1} \left(\prod_{i=1}^{t-1} K_\theta^i\right) \sigma_t\right) \mathbb{E}_{\boldsymbol{\epsilon},\boldsymbol{\epsilon}'} [\|\boldsymbol{\epsilon} - \boldsymbol{\epsilon}'\|]. \tag{14}$$

Using the induction hypothesis, the linearity of the expectation, Lemma 3.3 with $t := T$,

$$\mathbb{E}_{p_\theta(\mathbf{x}_{T-1}|\mathbf{x}_T) p_\theta(\mathbf{y}_{T-1}|\mathbf{y}_T)} \cdots \mathbb{E}_{p_\theta(\mathbf{x}_0|\mathbf{x}_1) p_\theta(\mathbf{y}_0|\mathbf{y}_1)} \|\mathbf{x}_0 - \mathbf{y}_0\|$$

$$\le \mathbb{E}_{p_\theta(\mathbf{x}_{T-1}|\mathbf{x}_T) p_\theta(\mathbf{y}_{T-1}|\mathbf{y}_T)} \left[\left(\prod_{t=1}^{T-1} K_\theta^t\right) \|\mathbf{x}_{T-1} - \mathbf{y}_{T-1}\| + \left(\sum_{t=2}^{T-1} \left(\prod_{i=1}^{t-1} K_\theta^i\right) \sigma_t\right) \mathbb{E}_{\boldsymbol{\epsilon},\boldsymbol{\epsilon}'} [\|\boldsymbol{\epsilon} - \boldsymbol{\epsilon}'\|]\right]$$

$$= \left(\prod_{t=1}^{T-1} K_\theta^t\right) \mathbb{E}_{p_\theta(\mathbf{x}_{T-1}|\mathbf{x}_T) p_\theta(\mathbf{y}_{T-1}|\mathbf{y}_T)} [\|\mathbf{x}_{T-1} - \mathbf{y}_{T-1}\|] + \left(\sum_{t=2}^{T-1} \left(\prod_{i=1}^{t-1} K_\theta^i\right) \sigma_t\right) \mathbb{E}_{\boldsymbol{\epsilon},\boldsymbol{\epsilon}'} [\|\boldsymbol{\epsilon} - \boldsymbol{\epsilon}'\|]$$

$$\le \left(\prod_{t=1}^{T-1} K_\theta^t\right) \left[K_\theta^t \|\mathbf{x}_T - \mathbf{y}_T\| + \sigma_T \mathbb{E}_{\boldsymbol{\epsilon},\boldsymbol{\epsilon}'} \|\boldsymbol{\epsilon} - \boldsymbol{\epsilon}'\|\right] + \left(\sum_{t=2}^{T-1} \left(\prod_{i=1}^{t-1} K_\theta^i\right) \sigma_t\right) \mathbb{E}_{\boldsymbol{\epsilon},\boldsymbol{\epsilon}'} [\|\boldsymbol{\epsilon} - \boldsymbol{\epsilon}'\|]$$

$$= \left(\prod_{t=1}^{T} K_\theta^t\right) \|\mathbf{x}_T - \mathbf{y}_T\| + \left[\left(\prod_{t=1}^{T-1} K_\theta^t\right) \sigma_T + \sum_{t=2}^{T-1} \left(\prod_{i=1}^{t-1} K_\theta^i\right) \sigma_t\right] \mathbb{E}_{\boldsymbol{\epsilon},\boldsymbol{\epsilon}'} [\|\boldsymbol{\epsilon} - \boldsymbol{\epsilon}'\|]$$

$$= \left(\prod_{t=1}^{T} K_\theta^t\right) \|\mathbf{x}_T - \mathbf{y}_T\| + \left[\left(\prod_{i=1}^{T-1} K_\theta^i\right) \sigma_T + \sum_{t=2}^{T-1} \left(\prod_{i=1}^{t-1} K_\theta^i\right) \sigma_t\right] \mathbb{E}_{\boldsymbol{\epsilon},\boldsymbol{\epsilon}'} [\|\boldsymbol{\epsilon} - \boldsymbol{\epsilon}'\|]$$

$$= \left(\prod_{t=1}^{T} K_\theta^t\right) \|\mathbf{x}_T - \mathbf{y}_T\| + \left(\sum_{t=2}^{T} \left(\prod_{i=1}^{t-1} K_\theta^i\right) \sigma_t\right) \mathbb{E}_{\boldsymbol{\epsilon},\boldsymbol{\epsilon}'} [\|\boldsymbol{\epsilon} - \boldsymbol{\epsilon}'\|].$$

$\square$

## B  Numerical Experiments

The goal of these experiments is to assess the numerical value of the bound of Theorem 3.1 on a synthetic dataset. The data-generating distribution is chosen to be the uniform distribution on the square of side 2, centered at the origin. Figure 2 shows samples from this target distribution.

The backward process uses a shared network with fully connected layers, and 128 hidden units each. The model is trained on $50,000$ samples from the original distribution, and the bound is computed with $n = 5,000$ independent samples. Samples from the trained model are shown in Figure 3.

We computed the bound for different values of $\lambda$. Given that the datapoints are confined to a square of side 2, using the primal form of the Wasserstein distance yields a straightforward upper bound of $W_1(\mu, \pi_\theta(\cdot)) \le \sqrt{8} \approx 2.828$. We estimated the Lipschitz norms $K_\theta^t$ using $K_t'$ from Remark 3.2, and the expected norms in the last two terms of Theorem 3.1 are estimated using $10^6$ independent samples from each distribution.

| $\lambda$ | $n/10$ | $n/5$ | $n/2$ | $n$ | $n/0.5$ | $n/0.1$ |
|---|---|---|---|---|---|---|
| Bound value | 1.124 | 1.231 | 1.5181 | 2.035 | 3.056 | 11.061 |

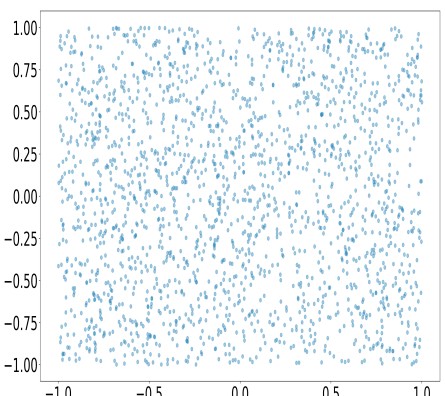

Figure 2: The points represent 2000 samples from the target data-generating distribution.

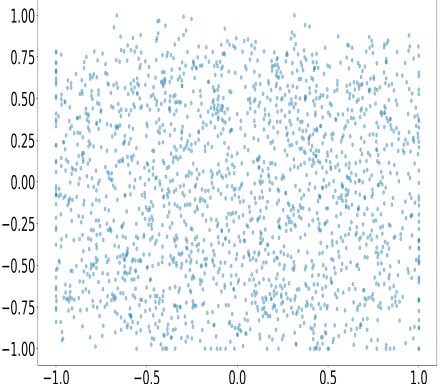

Figure 3: The points represent 2000 samples from the trained diffusion model.

