# OpenReview forum: "A Note on the Convergence of Denoising Diffusion Probabilistic Models"
_TMLR — Accepted by TMLR_

### Review · Reviewer_js5n · 2024-02-16

**Summary Of Contributions:**

This paper establishes a distribution estimation bound (stated in the Wasserstein distance) of diffusion models. Compared to existing sampling theory of diffusion models, the assumptions in the paper are mild and not imposed directly on the data distribution and the score functions. In this regard, the contribution of the paper is a new error bound of diffusion models, assuming the backward conditional mean is Lipschitz in its argument.

It is also to be noted that the error bound in the paper does not cover statistical estimation of the score function and the optimization guarantee of minimizing the score estimation loss. Therefore, the obtained result is not directly comparable to some recent advances in the statistical theory of diffusion models.

**Audience:**

Yes

**Claims And Evidence:**

No

**Requested Changes:**

1. $K_i$ in Theorem 3.1 Equation (5) seems undefined.

2. A particular choice of $\lambda \propto n^2$ needs further discussion. Substituting $\lambda \propto n^2$ will make Equation (5) grow linearly with $n$.

3. Following the second question, I would suggest providing explicit error rate implied by Theorem 3.1. Although an instantiation is provided just before Section 4, the convergence rate is still largely unclear. As the authors claim that the bound is obtained with weaker assumptions, how does the obtained bound compares to existing ones? I am also curious how the last error term $\sqrt{2D} (\sum (\prod K_i) \sigma_t)$ depends on $n$.

4. There seems to be a mismatch between Assumption 3.1 and Remark 3.2. The justification in Remark 3.2 needs a conditioning on the initial data point $x_0$, while Assumption 3.1 is a marginalized version. How does the justification circumvent this issue? In fact, if we consider the score-based diffusion models, Assumption 3.1 is almost equivalent to requiring the score function being Lipschitz continuous, as $g_\theta^t$ is determined by the score function (see Sampling is as easy as learning the score, by Chen et al., 2023).

**Strengths And Weaknesses:**

As far as I know, the obtained error bound of diffusion models is new in the literature. The terms in the error bound is relatively easy to understand, although some detailed discussion should be further provided.

The paper is also well organized, despite lacking more details on the preliminary of diffusion models. This might make unfamiliar readers struggle a bit.

The technical derivations are mostly correct and sound. However, there are some undefined notations (or typos) and one particular unjustified claim.

I will discuss weaknesses and raise questions in the next section.

---

> ### Author Response · Authors · 2024-03-29
>
> We thank the reviewer for reading our work, asking great questions and making thoughtful suggestions.
>
> > As far as I know, the obtained error bound of diffusion models is new in the literature. The terms in the error bound is relatively easy to understand, although some detailed discussion should be further provided.
>
> We thank the reviewer for the suggestions. We expanded the discussion under the main theorem a bit, and highlighted the tutorial of (Luo 2022) in the preliminaries.
>
> > $K_i$  in Theorem 3.1 Equation (5) seems undefined.
>
> This is a typo, it should be $K_\theta^i$. We corrected it in the new version of the paper, and we apologize for the confusion this may have caused.
>
> > A particular choice of $\lambda$ needs further discussion. Substituting $\lambda \propto n^2$ will make Equation (5) grow linearly with $n$.
>
> We assume the reviewer is referring to Equation (8), in Theorem 3.1. This is a typo, as we meant to highlight the fact that if $\lambda \propto n^{1/2}$, then the bound converges to a smaller quantity, since the term $\frac{\lambda\Delta^2}{8n}$ goes to $0$, but the convergence is slower. This is because the result of Mbacke et al. 2023 used in the paper stems from PAC-Bayesian theory, in which this particular trade-off is standard.
>
> > Following the second question, I would suggest providing explicit error rate implied by Theorem 3.1. Although an instantiation is provided just before Section 4, the convergence rate is still largely unclear. As the authors claim that the bound is obtained with weaker assumptions, how does the obtained bound compares to existing ones?
>
> As mentioned by the reviewer, "the obtained result is not directly comparable to some recent advances in the statistical theory of diffusion models." The convergence of our bound is not to $0$, but to the empirical loss computed using the training set. Bounds obtained in PAC-Bayesian theory guarantee low population risk (in this case the Wasserstein distance between $\mu$ and $\pi_\theta(\cdot)$) if the empirical risk is low and the additional terms are controlled properly. In that sense, we do not claim to have faster rates of convergence than the results in the literature, since the results are not directly comparable. Instead, our contributions are the following:
>
> * We remove the score estimation accuracy assumption (which is next to impossible to verify, as mentioned by Chen et al. 2023). To the best of our knowledge, ours is the first work to do this.
>
> * We derive quantitative, Wasserstein distance based upper bounds, which is also novel;
>
> * We approach the problem from a different perspective (DDPMs instead of SGMs), and use elementary proof techniques.
>
> Finally, as requested by another reviewer, we added numerical experiments on a synthetic dataset.
>
> > I am also curious how the last error term $\sqrt{2D} (\sum (\prod K_\theta^t )\sigma_t)$ depends on  $n$.
>
> This term does not depend on $n$, although it does depend on number of timesteps $T$ and the ambient dimension $D$. The term is obtained using a standard upper bound on the average distance between Gaussian vectors.
>
> > There seems to be a mismatch between Assumption 3.1 and Remark 3.2. The justification in Remark 3.2 needs a conditioning on the initial data point $x_0$, while Assumption 3.1 is a marginalized version. How does the justification circumvent this issue?
>
> This is a great question. The justification in Remark 3.2 shows that the Lipschitz norm of the mean function does *not* depend on $x_0$. Indeed, looking at the equation just below Equation (11), we can see that for any initial $x_0$, the Lipschitz norm
> ($K_t' = \frac{\sqrt{\alpha_t} (1-\bar{\alpha}_{t-1})}{1-\bar{\alpha}_t}$)
>
> only depends on the noise schedule, not $x_0$ itself. Now, as the reviewer knows, $g_\theta^t$ is optimized to match to $\mu_q(\cdot, x_0)$, for each $x_0$ in the training set. Since, as shown in Remark 3.2, all the functions $\mu_q(\cdot, x_0)$ have the same Lipschitz norm
> $\frac{\sqrt{\alpha_t} (1-\bar{\alpha}_{t-1})}{1-\bar{\alpha}_t}$,
>
> we believe it is reasonable to use this Lipschitz norm to get a sense of the scale of the Lipschitz norm of $g_\theta^t$.
>
> >  Assumption 3.1 is almost equivalent to requiring the score function being Lipschitz continuous, as $g_\theta^t$ is determined by the score function (see Sampling is as easy as learning the score, by Chen et al., 2023).
>
> Indeed the score function of the backward process is dependent on $g_\theta^t$. In contrast to our assumption on the backward process, Chen et al., 2023 assume the forward process is Lipschitz ((A1) in that paper), and they also assume an upper bound on the score estimation error. Note that we do not make such an assumption, and our result does not even require the true distribution to have a score function.
>
> Once again, we thank the reviewer for their hard work and suggestions, and we are happy to answer any further questions.

---

### Review · Reviewer_3yzG · 2024-02-27

**Summary Of Contributions:**

The authors provide a bound on the approximation quality of the learned distribution in a diffusion model. The proof technique appears to follow closely to a prior work on the approximation quality of variational autoencoders [1].

[1] Statistical Guarantees for Variational Autoencoders using PAC-Bayesian Theory. https://arxiv.org/abs/2310.04935

**Audience:**

Yes

**Broader Impact Concerns:**

No concerns.

**Claims And Evidence:**

Yes

**Requested Changes:**

I am interested in a simple synthetic experiment (e.g. on a 1D Gaussian data-generating distribution) that demonstrates the tightness of the proposed bound (perhaps at different $\delta$). Is the bound relatively tight? Or is it a loose bound?

If there are empirical observations that can help guide the design and training of diffusion models, can the author elaborate on them?

**Strengths And Weaknesses:**

Strengths:
- The theorem relaxes some assumptions present in previous theoretical results, including those on the score function and data-generating distribution.
- Diffusion models are a highly relevant topic.

Weaknesses:
- As I understand it, the tightness of the bound is inversely related to the likelihood of the bound being true (as controlled by the $\delta$ term). This is inherited from the underlying PAC-Bayes bound for bounded loss functions.
- The present manuscript does not provide any numerical experimentation demonstrating the tightness of this bound.
- To my understanding, there are limited empirical takeaways from this result, despite its nice theoretical properties. If I am missing something, I would appreciate that the authors clarify on this point.

---

> ### Author Response · Authors · 2024-03-29
>
> We thank the reviewer for reviewing our work and making great suggestions.
>
> > As I understand it, the tightness of the bound is inversely related to the likelihood of the bound being true (as controlled by the $\delta$ term). This is inherited from the underlying PAC-Bayes bound for bounded loss functions.
>
> This observation is correct, meaning that $\delta$ impacts tightness of the bound in the sense that higher confidence (smaller $\delta$) goes with looser bound. This is not inherent to PAC-Bayes bounds but PAC bounds in general (e.g. the classical Hoeffding bound). Typically the choice of $\delta$ is problem-dependent, and comes down to what sort of confidence one wishes to guarantee. However, the real difference between "tight" and "loose" comes from other terms of the bound, e.g. dependence on sample size and the hyperparameter $\lambda$.
>
> > To my understanding, there are limited empirical takeaways from this result, despite its nice theoretical properties.
>
> Indeed, the contribution of the paper is theoretical, although we added some numerical experiments on a synthetic dataset. See below.
>
> > I am interested in a simple synthetic experiment (e.g. on a 1D Gaussian data-generating distribution) that demonstrates the tightness of the proposed bound (perhaps at different ). Is the bound relatively tight? Or is it a loose bound?
>
> We performed experiments with the following simple 2D dataset: the uniform distribution on the square of side $2$, centered at the origin. We trained a model, then computed the bound of Theorem 3.1 for different values of $\lambda$. Note that we chose different values of $\lambda$ because it has a much higher incidence on the numerical values of the bound than $\delta$.
>
> For this particular dataset, since the side of the square is $2$, a straightforward upper bound on the Wasserstein distance between the two distribution is $\sqrt{8} \approx 2.828$. The numerical value of the bound was around  1.5, depending on the choice of the hyperparameter $\lambda$. We added the full details in the supplementary material.
>
> > If there are empirical observations that can help guide the design and training of diffusion models, can the author elaborate on them?
>
> The bound provides an upper bound for a trained  model, not a new training objective. In this sense, the primary objective of this work was to derive a guarantee for the distribution learned by the diffusion model. However, PAC-Bayes bounds have been used in the past to develop novel algorithms, so we leave these investigations to future works.
>
> Once again we thank the reviewer for their hard work and thoughtful comments.

---

### Review · Reviewer_ckGf · 2024-03-25

**Summary Of Contributions:**

This paper provides upper bounds on the Wasserstein distance between the true distribution and the learned distribution of the diffusion denoising framework. These upper bounds do not require any assumptions on the data distribution or the score function.

**Audience:**

Yes

**Claims And Evidence:**

Yes

**Requested Changes:**

Addressing the above concerns will go a long way in making this paper useful to the community.

Addition of some illustrative examples where the proposed bounds imply consistency or optimality of the learned distribution would be very helpful. Moreover, contexts in which the provided bounds are applicable to real world data distribution would be useful. Do the bounds depend on the ambient dimension of the data distribution or the implicit dimension? I would guess the dependence on the dimension would still be exponential but for what other parameters of the problem is it not? Is this the defining difference of the results of this paper as compared to the related work?

**Strengths And Weaknesses:**

The bounds provided on the Wasserstein distance between the data distribution and the learned distribution are
- novel.
- do not require any assumptions on the data distribution
- do not require any assumptions on the score function of the data distribution

Weaknesses:
- The claims of the paper are ambiguous and misleading at times:
——For eg., it is not clear “no exponential dependencies” is referring to. Exponential in what parameters? The paper also claims that the results are different from others by providing “quantitative” upper bounds. It is unclear what this is referring to.
- The upper bounds will of course be exponential in the dimension.
- The upper bounds are provided in a crude form. It is not clear from this expression when these bounds would be non-trivial. This greatly impacts the usefulness of this paper to the community. It’s not evident where these bounds fit in the context of other density estimation bounds in the literature.
- In fact, it is not even clear if the bounds imply consistency let alone optimality of the estimator provided by the diffusion model.

Minor Issues:
- The definition of the distribution \pi_\theta(.) is not illustrative or illuminating.
- Is there a reason we bound the upper bound with the distance from the regenerated empirical distribution?
- Why is the assumption of the Lipschitz of the functions g_\theta reasonable?

---

> ### Author Response · Authors · 2024-03-29
>
> We thank the reviewer for reading and asking thoughtful questions.
>
> > For eg., it is not clear “no exponential dependencies” is referring to. Exponential in what parameters? ... The upper bounds will of course be exponential in the dimension.
>
> This refers to the fact that the terms in the right-hand side of the bound do not exponentially depend on the parameters of the problem. Looking at the last inequality before the conclusion, we can see that the dependency in the dimension $D$ of the ambient space is polynomial, not exponential. Note that this is not "too good to be true" since there are drawbacks, as we mentioned in the paper and highlight below.
>
> > The paper also claims that the results are different from others by providing “quantitative” upper bounds. It is unclear what this is referring to.
>
> This is referring to the fact that the bounds are explicit quantities, as opposed to just rates of convergence. So, instead of having an upper bound that is, for instance $O(1/n)$ with hidden constants, we have an upper bound where the expressions are explicitly given, and can be computed or estimated numerically. On the other hand, if the result is given in terms of rate of convergence, then the upper bound cannot be computed numerically because the exact expression is unknown. Sometimes, the term "empirical upper bound" is also used in the literature. We used "quantitative" because it's fairly standard, see eg Chen et al. 2023, where the authors derive quantitative total-variation based bounds for SGMs.
>
> > The upper bounds are provided in a crude form. It is not clear from this expression when these bounds would be non-trivial.
>
> Admittedly yes. Theorem 3.1 is interesting because it is the first bound of its kind for diffusion models, but we make no claims about the sharpness of this inequality. It is possible that some other arguments / proof techniques can deliver a sharper inequality, but again we think ours is interesting nonetheless and it could motivate interest in deriving sharper inequalities in the future. Moreover, the experiments we performed on a 2D dataset (see appendix) show that the bound can be non-vacuous for synthetic datasets.
>
> > In fact, it is not even clear if the bounds imply consistency let alone optimality of the estimator provided by the diffusion model.
>
> The bound depends on the performance of the model on a finite i.i.d. sample, so it shows that the lower the risk on that sample is, the closer the learned distribution is to the true distribution. We are not sure what the reviewer means by optimality, but the convergence is one of the drawbacks, since the last term does not depend on $n$.
>
> > The definition of the distribution $\pi_\theta(.)$ is not illustrative or illuminating.
>
> We understand the definition might be a bit unclear. This is why we also added the definition in terms of expectations. It is the distribution of the samples obtained by sampling first from the prior  $p(x_T)$, and passing through the entire backwards process.  In summary, $\pi_\theta(\cdot)$ is the distribution learned by the diffusion model. We hope this makes things clearer.
>
> > Is there a reason we bound the upper bound with the distance from the regenerated empirical distribution?
>
> Bounding the distance between the learned and true distributions directly is challenging. So the regenerated distribution is the intermediate distribution we used to solve this problem. Other works used other intermediate distributions, (see e.g. Equation (8) of De Bortoli (2022) TMLR).
>
> > Why is the assumption of the Lipschitz of the functions $g_\theta$ reasonable?
>
> We believe this assumption to be reasonable because $g_\theta^t$ is trained to match $x_t \mapsto \mu_q(x_t, x_0)$, which is the mean of the distribution $q(x_{t-1} | x_t , x_0)$ and, as shown in Remark 3.2, $\mu_q(\cdot, x_0)$ is Lipschitz. Moreover, if $(\alpha_t)_{1 \leq t \leq T}$ is a decreasing sequence less than $1$ (which is what's generally used in practice), then the Lipschitz norm of $\mu_q(\cdot, x_0)$ is $< 1$. This is why we believe that assumption to be reasonable.
>
> Once again, we thank the reviewer for their hard work, and we will happily answer any further questions.

---

> > ### Comment · Reviewer_ckGf · 2024-04-29
> > **Polynomial dependence on dimension.**
> >
> > I'm not sure I follow why polynomial dependence on the dimension is not "too good to be true". Can you clarify?

---

> > > ### Author Response · Authors · 2024-04-30
> > >
> > > We thank the reviewer for their response.
> > >
> > > Our comment on the polynomial dependence on the dimension refers to the fact that, as far as we know, the existing upper bounds for diffusion models in the literature have exponential dependencies on the dimension. Our bound does not suffer from this, but it holds with high probability on the randomness of the observed samples (whereas existing bounds hold with probability one), and the dimension still affects any numerical computation of the bound, since the diameter of the instance space depends on the dimension, if we consider a standard metric such as the Euclidean distance.
> > >
> > > We hope this answers the reviewer's question. We are happy to provide any further clarifications.

---

### Review · Reviewer_PJMq · 2024-04-17

**Summary Of Contributions:**

This work proposes a bound on the Wasserstein distance between the parameterized distribution and the true data distribution

**Audience:**

Yes

**Claims And Evidence:**

Yes

**Requested Changes:**

I am sorry for submitting the review so late. While I think the work can improve on the above three points I mentioned. I have no serious objection to the paper.

Perhaps the main thing to clarify and add to the work is to clarify why the theory is novel -- given that it seems standard to apply it to VAEs.

**Strengths And Weaknesses:**

Strength:
1. the problem is fundamental and interesting to study per se

Weakness:
1. The usefulness of the theory is unclear
2. Lack of numerical evidence
3. The theory seems to be standard (?) in the field of the study of VAEs, is this the case?

---

> ### Author Response · Authors · 2024-04-27
>
> We thank the reviewer for their hard work and thoughtful comments. We answer the reviewer's three questions below.
>
> > The usefulness of the theory is unclear.
>
> Here are the reasons why we believe the theory results presented in our paper to be worthy of attention. We use a different approach from the existing literature; we avoid assumptions on the score function; the bound is quantitative, as opposed to rates of convergence; and we also avoid exponential dependencies. The proofs are based on elementary techniques, which we consider to be an advantage because this allows more researchers to understand and further build on these results. Also, Theorem 3.1 is interesting because it is the first bound of its kind for diffusion models (cf. our response to Reviewer ckGf), and, as commented by Reviewer js5n, "As far as I know, the obtained error bound of diffusion models is new in the literature."
>
> > Lack of numerical evidence
>
> As requested by another reviewer (3yzG), we ran some experiments on a synthetic dataset and reported the experimental details, as well as the results, in the appendix of the updated paper. The code is also available in the supplementary material.
>
> > The theory seems to be standard (?) in the field of the study of VAEs, is this the case?
>
> The bound is indeed based on the work of Mbacke et al. (NeurIPS, 2023), which studies  VAEs.. Although as far as we are aware, other than the recent Mbacke et al. 2023, there are no other previous works in the literature proposing this kind of theory for VAEs.
>
> Once again, we thank the reviewer for evaluating our work. We are happy to answer any further questions.

---

### Decision · Action_Editor_enWw · 2024-05-29

**Recommendation:** Accept with minor revision

**Comment:**

Reviewer js5n has raised multiple concerns and authors addresses most of those in their rebuttal. However, I ask authors to carefully go over the suggested items by this reviewer and make another attempt to address any remaining concerns; in particular, see the reviewers comment 3 and 4.

**Audience:**

The subject is of interest to the TMLR audience.

**Claims And Evidence:**

Yes, the reviewers found that the technical derivations are correct and sound.